# The impact of Internet use and involvement on residents' attitudes to healthcare in China: A propensity score matching analysis

Xinyue Li[1], Song Zhang[2], Xiaokang Song[2]*

1 School of Information Management, Nanjing University, Nanjing, Jiangsu, China, 2 School of Management, Xuzhou Medical University, Xuzhou, Jiangsu, China

* sxksxk666@163.com

## Abstract

The recent surge in Internet growth has significantly altered how residents obtain health information and services, underscoring the need to investigate its impact on healthcare perceptions. However, current studies often fail to distinguish between Internet use and involvement, as well as the diverse range of healthcare stakeholders, resulting in incomplete and inconsistent understanding. To address this, this study utilized data from the 2018 China Family Panel Study (CFPS 2018), categorizing attitudes toward healthcare into three dimensions: doctor trust, satisfaction with medical institutions, and perception of systemic healthcare issues. Employing propensity score matching (PSM) to control for thirteen confounding variables, this study examined the Internet's impact on public attitudes toward healthcare among similar demographic, psychological, and health-related variables. Results revealed that both Internet use and involvement affect residents' attitudes toward healthcare to some extent, with involvement having a more pronounced effect. While Internet use increased the perception of systemic healthcare issues, Internet involvement enhanced doctor trust, yet reduced satisfaction with medical institutions and exaggerated the perception of systemic healthcare issues. These findings have significant theoretical and practical implications. They enhance the comprehension of diverse levels and purposes of Internet use, thereby advancing our knowledge of its multi-faced influence on public attitudes toward healthcare. Furthermore, they offer insights for medical institutions to improve service quality, assist Internet media in optimizing information delivery, and illuminate the implications for residents who effectively use the Internet to assess health information.

## Introduction

China, as one of the world's largest online markets, has seen a rapid expansion of the Internet alongside an uneven distribution of medical resources between urban and rural areas. This disparity has led to a growing reliance on the Internet as the primary source of health information for many residents. Digital platforms, including search engines, online health communities, Q&A forums, and social media, have become pivotal in providing access to

**Data Availability Statement:** The data that support this study are available from the [Institute of Social Science Survey, Peking University] but restrictions apply to the availability of these data, which were used under license for the current study. In

addition, the processed data are contained in supporting information files.

**Funding:** This work was supported by the National Natural Science Foundation of China under Grant [number 72204210] and the Scientific Research Foundation of Xuzhou Medical University under Grant [number D2021039].

**Competing interests:** The authors have declared that no competing interests exist.

**Abbreviations:** ATT, Average treatment effect on the treated; CFPS, China Family Panel Study; PSM, Propensity score matching.

health information, with their convenience and accessibility enabling users to make appointments (63%), access medical advice (47%), purchase medications (37%), manage the disease (10%), and even consult with professionals remotely (16%) [1]. The COVID-19 pandemic has accelerated the adoption of Internet healthcare services in China, resulting in the establishment of 2,700 internet hospitals and a substantial increase in online medical consultations, which have soared to 25.9 million instances [2]. It is predicted that the market for Internet hospitals will continue to expand, reaching a penetration rate of 30% and a market size of 500 billion by 2025 [3]. According to the 53rd Statistical Report of the China Internet Network Information Center (CNNIC), by December 2023, over 414 million Internet users in China were engaged in accessing healthcare-related information, accounting for 37.9% of all Internet users [4]. While online healthcare offers benefits such as reducing disparities in medical access and delivering up-to-date information [5], it also poses risks, including the spread of health information and online disparities [6]. Consequently, it is crucial to explore the impact of the Internet on residents' attitudes toward healthcare in China.

The Internet exerts a tangible influence on individuals' attitudes to health, their decision-making, and their relationship with the healthcare system [7, 8]. Nonetheless, extant research has yielded inconsistent findings regarding the impact of the Internet on residents' attitudes toward healthcare. Some studies highlighted the potential benefits such as facilitating communication, and bridging the information asymmetry between doctors and patients, thereby improving doctor-patient relationships, and assisting individual health-related decision-making [9]. On the other hand, massively misleading and misinterpreted information, the inconsistency of online sources with doctors' diagnoses, and residents' low health e-literacy could lead to doctor-patient conflict and the misunderstanding of healthcare information [10]. This is compounded by negative bias theory, which holds that individuals tend to pay more attention to negative than positive information [11, 12]. To rectify disparities in prior findings and unveil the genuine impact of the Internet on the residents' attitudes toward healthcare, this study scrutinizes specific differences in Internet use and involvement, categorizing public attitudes into three distinct dimensions. Employing propensity score matching to mitigate covariate selection bias, this study leverages large-scale secondary data to derive more broadly applicable conclusions. Overall, comprehending the influence mechanism will facilitate a more effective utilization of the Internet's potential within the healthcare domain shortly.

## Literature review

### Internet use and Internet involvement

Internet use, defined as accessing and utilizing the Internet for information obtaining and decision-making [13], was initially understood as a novel form of media engagement [14]. For nearly two decades, its impact has been a central focus in information behavior research. However, the rapid advancement in information technology and the enhancement of public digital literacy have outpaced this construct's ability to fully capture how the public engages with the Internet, leading to the emergence of Internet involvement. Beyond mere usage, involvement delves into the significance and personal relevance users attribute to the Internet, which varies based on their focus [15] (Table 1). This evolution aligns with the societal and academic focus from the first-level digital divide of "access" [16] to addressing the second-level digital divide of "usage/utility" [17]. Examining Internet access can showcase technological spread, but it inadequately addresses the broader social effects of this diffusion [18]. Moreover, variations in Internet involvement, influenced by different digital skills and purposes, can significantly affect individual attitudes and behaviors [19]. Therefore, concentrating only on the binary

**Table 1. Comparative analysis of Internet use and Internet involvement.**

|  | Definition | Attribute | Measurement | Corresponding Digital Divide |
|---|---|---|---|---|
| Internet Use | Simple access and utilization of the Internet (i.e., PC, mobile phone) to obtain information and make decisions [13] | Physical [20] | Whether or not to use | First-level divide: access [16] |
| Internet Involvement | Importance/Relevance of using the Internet to obtain information and make decisions [15] | Psychological [20] | How much an individual uses or values the Internet | Second-level divide: usage/ utility [17] |

perspective of "having or not having" Internet access is insufficient for fully grasping its influence on residents' attitudes toward healthcare.

The impact of the Internet on the public's attitudes to healthcare has long been a concern. However, previous research has shown considerable discrepancies, likely due to the different interpretations of "Internet use", a viewpoint supported by Wang and Meng [21]. Traditionally, studies have treated Internet use as a binary concept, focusing solely on the impact of an individual's decision of whether or not to use it on their attitudes. For instance, Ybarra and Suman, through secondary data from Surveying the Digital Future, examined how accessing health information online could shape individuals' healthcare experiences [9]. However, evidence suggests that the impact of Internet use is multi-faced and varies with engagement levels and purposes [22]. Yuan et al. differentiated between Internet use and involvement to explore their distinct effects on the cognitive health of middle-aged [23]. Nguyen et al. discovered that despite widespread Internet access in California, disparities in online health information-seeking persist, influencing public attitudes differently [24]. This highlights the importance of differentiating between Internet use and Internet engagement to fully understand its impact on public attitudes toward healthcare, as suggested by the concept of the "first- and second-level digital divide" [25]. Given the gap in empirical research, this study aims to compare the effects of Internet use and involvement on healthcare attitudes in China, where digital disparities are significant.

## Residents' attitudes toward healthcare

Residents' attitudes toward healthcare profoundly affect their health outcomes [9] and the evolution of healthcare services in China [26]. The landscape of healthcare provision involves a broad range of stakeholders, including doctors, regulators, and health-related companies [27]. Doctor trust, rooted in patient expectations and beliefs about their physicians, is central to their relationship [28]. This trust is pivotal for effective health management and enhancing patient satisfaction [29]. However, trust is a complicated notion that is shaped by various factors such as perceived comfort, personal involvement, or even doctors' appearance in different social contexts [30]. Recent scholarship has intriguingly explored the role of trust in doctors in the context of increasing Internet interventions in healthcare [31].

Compared to their viewpoints on doctors, residents generally harbor more negative and distrustful attitudes towards medical institutions and their issues. A survey of UK consumers found that the public generally showed low trust in healthcare institutions, contrasting with high trust in doctors [32]. Since the 1990s, China's healthcare system has been grappling with a trust crisis among its residents [26]. Zhao et al. highlighted low public trust in China's healthcare as a critical concern [33], potentially fueled by underlying issues such as medical corruption, over-medication, and drug unreliability [34]. Public attitudes towards healthcare services are crucial, yet research on how the Internet influences them is limited, especially under the "Internet + healthcare" policy. Most studies have focused on doctor-patient relationships, leaving a gap in understanding public viewpoints on other healthcare stakeholders. This study

aims to fill that gap by exploring residents' attitudes toward various healthcare entities, providing a more nuanced understanding.

## Theoretical analysis

To gain deeper insights into the Internet and attitudes toward healthcare services, this study develops a theoretical analysis. The Internet's influence, as a new technology and medium, is often analyzed through two lenses: the technology acceptance model [35] and the media bias theory [36]. The former posits that it enhances attitudes towards healthcare by improving accessibility and availability of health information, and the latter argues that the information can be skewed or distorted online, resulting in heightened negative emotions. This study posits that the challenge in consolidating perspectives on the Internet's influence likely stems from the failure of scholars to distinguish Internet use and involvement and to appropriately segment attitudes across multiple dimensions.

Given the widespread impact of Internet integration across healthcare, affecting individuals, providers, organizations, and communities [37], attitudes toward healthcare should be divided into distinct dimensions, focusing on doctors, institutions, and issues. First, regarding doctor trust, scholars present two conflicting perspectives on how Internet use affects it. The emergence of the Internet, granting residents broader access to health information and insights, has transformed the nature of doctor trust from "blind trust" to "informed trust" [38]. Soroka et al. argued that the Internet's flexibility and interactivity can promote mutual communication between doctors and patients, which in turn enhances doctor trust [39]. Similarly, Akerkar and Bichile suggest the Internet offers an excellent avenue to augment doctor trust by redistributing the responsibility for knowledge sharing [38]. However, other scholars, like Murray et al., discovered that the relationship suffers when patients encounter inaccurate or irrelevant information online, often perceived by doctors as a challenge to their authority [40]. Pham et al. also observed a decline in doctor trust when online health information contradicted a doctor's advice, leading individuals to seek guidance from alternative sources [41].

Second, regarding attitudes toward institutions, satisfaction is traditionally seen as a key indicator of the quality of healthcare institutions [40]. Despite the advancements in the quality and accessibility of online healthcare services facilitated by China's "Internet + Healthcare" initiative, Liu et al. discovered that Internet use might intensify individual dissatisfaction with healthcare services, aligning with the negative bias theory [42]. The perpetuation of scandals and adverse reports online contribute to a growing discontent with healthcare systems [43]. Finally, when it comes to attitudes toward issues, systemic healthcare issue perception encapsulates the general viewpoint residents have on matters related to healthcare [26]. Nonetheless, the extensive, often negative, information available on the Internet can lead to the "Negativity Effect" among individuals regarding public issues [39].

Moreover, this study suggests that varying levels of Internet usage have differential effects on the residents regarding various healthcare subjects. On one hand, mere Internet use implies basic access, while Internet involvement offers extensive information, fostering social participation and self-efficacy [44], potentially transforming their roles and interactions within the healthcare context. On the other hand, Internet use and involvement result in varied exposures to diverse healthcare subjects. Passive Internet use, exemplified by simple browsing, can readily introduce individuals to medical reports, consequently shifting their views on healthcare issues. In contrast, active Internet involvement, like communication and consultation in online health communities, can precisely shape the perceptions of healthcare professionals and institutions [45, 46].

However, the specific mechanisms behind how the Internet influences residents' multifaced attitudes toward healthcare remain a "black box". In addition, most existing literature

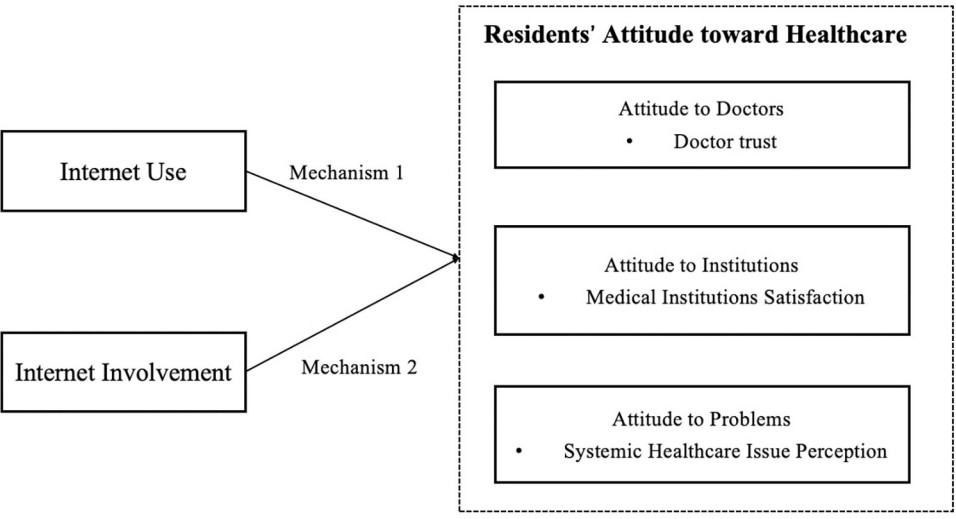

**Fig 1. Research framework.**

researched the correlation (rather than the casual relationship) between Internet use and attitudes to healthcare, by adopting a statistical method such as regression analysis, and ignored the impact of many confounding factors [47].To overcome the problems outlined previously, this study used propensity score matching (PSM) to explore the impact of Internet use and involvement on residents' attitudes to healthcare in China. This study examined two mechanisms, as shown in Fig 1. The first focused on whether Internet use impacted attitudes to healthcare, and the second on the effect of Internet involvement. The study divided public attitudes to healthcare into three categories: doctors, medical institutions, and systemic healthcare issues.

## Method

### Data and sampling

The data used in this study were from the 2018 China Family Panel Studies (CFPS 2018). Initiated by Peking University, the CFPS is a large-scale, comprehensive, and high-quality biennial social tracking survey. All participants in the survey were asked for written informed consent. The data were released to the researchers without access to any personal data. Collecting data at the individual, family, and community levels, reflected changes in demographic, educational, economic, health, and social spheres, and provided support for academic research [48]. The present study adopted demographic, Internet-related, and healthcare-related data at the individual level from CFPS 2018. After eliminating invalid entries (such as those missing Internet use or healthcare values), 25121 valid samples were obtained. The flow of sample selection is shown in Fig 2.

### Measurement

**Independent variables.** The CFPS 2018 contains three questions related to Internet use: (1) Do you use mobile devices to surf the Internet? (Yes/No) (2) Do you use a computer to surf the Internet? (Yes/No) (3) How important is the Internet as an information source? (Five-point scale) Since this study aims not only to explore the impact of Internet use on attitudes to healthcare but also to further unearth the extent of Internet involvement and its effects, it identified two independent variables. First, it considered respondents who answered "yes" to at

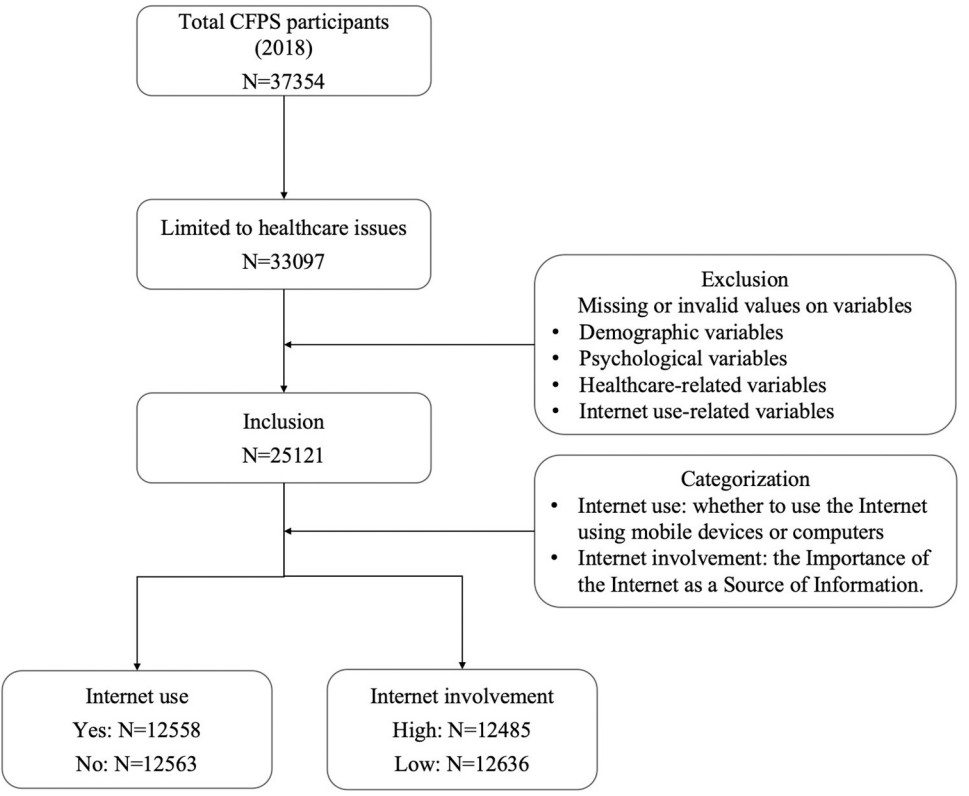

**Fig 2. Flow diagram of sample selections.**

least one of the first two questions as Internet users (Internet use: 1 = yes and 0 = no). Second, it regarded those who rated their use > 3 in question (3) as individuals with high Internet involvement, and those who rated it < 3 as individuals with low Internet involvement (Internet involvement: 1 = high and 0 = low). The study believed that question (3) could be used to explore the relationship between Internet involvement and attitudes to healthcare for two reasons. First, there are various forms of Internet use, such as playing games and chatting, and accessing information is a much more in-depth form. Second, given the development of Internet-based healthcare and the importance individuals place on health issues, if the public regards the Internet as a major information source the demand for online healthcare will increase accordingly.

**Dependent variables.** An individual's attitude to healthcare has three dimensions: doctors, medical institutions, and systemic healthcare issues. These can be broken down into doctor trust, medical institution satisfaction, and systemic healthcare issue perception, respectively. In the CFPS questionnaire, doctor trust was measured by a single question, with responses given on a scale from 1–10: "How much do you trust your doctor?" Medical institution satisfaction was measured by the question "How satisfied are you with the medical institution you visit?" Responses were given on a five-point scale: 5 ("Very unsatisfactory"), 4 ("Unsatisfactory"), 3 ("Fair"), 2 ("Satisfactory"), 1 ("Very satisfactory"). Systemic healthcare issues perception was measured by the question "How serious do you think China's systemic healthcare issues are?" Responses were given on a 10-point scale.

**Confounding covariates.** This study used thirteen confounding variables for internet use and attitudes to healthcare. These included demographic variables, such as age, education

**Table 2. Covariate definitions/codes.**

| Covariates | Definition/codes |
|---|---|
| Gender | Binary variable (1 = male; 0 = female) |
| Age | Continuous variable |
| Education level | Categorical variable |
| Marital status | Categorical variable (1 = unmarried; 2 = divorced; 3 = widowed; 4 = cohabiting; 5 = married) |
| Urban or rural | Binary variable (1 = urban; 0 = rural) |
| Not lonely | From 1 to 5 (not important to very important) |
| Life satisfaction | From 1 to 5 (very unsatisfactory to very satisfactory) |
| Health status | From 1 to 5 (very healthy to unhealthy) |
| Alcohol drinker | Binary variable (1 = yes; 0 = no) |
| Chronic disease | Binary variable (1 = yes; 0 = no) |
| Medical insurance | Binary variable (1 = yes; 0 = no) |
| Hospital admissions | Binary variable (1 = yes; 0 = no) |
| Hospital quality | From 1 to 5 (very bad to very good) |

level, gender, marital status, and urban or rural residence [49]. Previous studies highlight the potential relationship between emotional state and health information behavior [50], so this study used loneliness and life satisfaction as psychological variables. It also included health status, chronic diseases, and hospital admissions, to ascertain the effect of healthcare-related variables on respondents' attitudes to healthcare [51, 52]. A more detailed description of these confounding covariates is shown in Table 2.

## Statistical analysis

This study used Stata 16.0 as its main data analysis tool. First, it conducted a descriptive analysis to demonstrate the basic characteristics of the variables. It then used propensity score matching (PSM) as the key statistical technique to address the research questions. PSM was proposed by Rosenbaum and Rubin, and involves matching samples by calculating propensity scores for a series of covariates. PSM, which matches experimental group samples with natural case samples under similar conditions and compares their difference, is an effective method for mitigating covariate selection bias [53].

In the PSM analysis, the study first built a logistic regression model to calculate all respondents' propensity scores for Internet use and involvement, based on the thirteen confounding covariates selected. The propensity score was then used to match samples. The study also checked the covariate balancing between the two groups. Standardized bias is an appropriate metric with which to access the distance of marginal distribution of covariates, and a result of no more than 10% after matching indicates that PSM has balanced the effects of the covariates on both experimental and control groups. Next, matching was conducted based on the propensity score, to estimate the average treatment effect on the treated (ATT). In the formula for calculating ATT, $D_i$ is the observed variable in the study, for which 1 indicates an experimental group respondent, and 0 indicates a control group respondent:

$$ATT_{PSM} = E\{Y_{1i} - Y_{0i}|D_i = 1\} = E\{E[Y_{1i} - Y_{0i}|D_i = 1], P(X_i)\}$$
$$= E\{E[Y_{1i}|D_i = 1, P(X_i)] - E[Y_{0i}|D_i = 0, P(X_i)]|D_i = 1\}$$

As previously stated, this study focused on two mechanisms. In the first, respondents in the experimental group were Internet users and those in the control group were not. ATT was the average difference in doctor trust, institution satisfaction, and medical issue perceptions if all

treated residents used the Internet, compared to the same individuals had they not used it. In the second, the experimental group contained respondents with high levels of Internet involvement, while the control group had low levels. Finally, the study ensured the reliability of results through a balance test and sensitivity analysis.

## Results

### Descriptive statistics

Table 3 shows the respondents' characteristics. This study used 25121 samples, among which the ratio of men to women was almost balanced. The average age was 47.213 years, with a range of 16 to 96. Most respondents' education level was not high, most lived in rural areas, and most were married. Their health status was fair, and few had had a recent chronic disease. Over 90% of respondents had medical insurance. Their perceptions of happiness and lack of loneliness were relatively high, with mean scores above 4. Respondents generally had a high level of trust in doctors, with a rating of 6.7 or above, but only a fair level of satisfaction with medical institutions, and high levels of concern about systemic healthcare issues in China.

### Propensity score

This study conducted logistic regression models to calculate respondents' propensity scores for Internet use and Internet involvement, based on the thirteen covariates in Table 4. The logistic regression model is the model most frequently used to calculate propensity scores since it does not contain requirements for normal distribution or type of covariates [54]. Overall, the explanatory power of the two models was relatively good, and many covariates influenced Internet use and Internet involvement. Specifically, demographic variables like age, education level, and living in rural or urban areas had statistical significance in both Internet use and Internet involvement. Loneliness and life satisfaction also had a significant impact on

**Table 3. Demographics, baseline characteristics of respondents before matching.**

| Characteristics | Total | Internet use | | Internet involvement | |
|---|---|---|---|---|---|
| | N = 25121 | Yes | No | High | Low |
| | | N = 12563 | N = 12558 | N = 12485 | N = 12636 |
| Mean (SD) | | | | | |
| Gender | 0.492(0.499) | 0.515(0.499) | 0.469(0.499) | 0.514(0.500) | 0.469(0.499) |
| Age | 47.213(17.014) | 36.254(13.038) | 58.177(12.988) | 36.755(13.548) | 57.547(13.391) |
| Education level | 2.796(1.453) | 3.622(1.333) | 1.968(1.037) | 3.597(1.356) | 2.003(1.056) |
| Marital status | 2.034(0.836) | 1.788(0.645) | 2.280(0.927) | 1.791(0.655) | 2.274(0.921) |
| Urban or rural | 0.295(1.779) | 1.704(1.809) | 1.405(1.736) | 1.685(1.811) | 1.425(1.739) |
| Not lonely | 4.192(1.034) | 4.185(0.974) | 4.200(1.091) | 4.218(0.953) | 4.168(1.108) |
| Life satisfaction | 4.029(0.964) | 3.919(0.911) | 4.139(1.003) | 3.599(0.795) | 4.101(1.022) |
| Health status | 3.041(1.228) | 2.754(1.082) | 3.329(1.296) | 2.754(1.090) | 3.325(1.290) |
| Alcohol drinker | 0.149(0.356) | 0.134(0.341) | 0.164(0.371) | 0.137(0.344) | 0.161(0.367) |
| Chronic disease | 0.171(0.376) | 0.105(0.307) | 0.237(0.425) | 0.106(0.308) | 0.235(0.424) |
| Medical insurance | 0.915(0.278) | 0.906(0.293) | 0.925(0.263) | 7.168(2.486) | 0.925(0.262) |
| Hospital admissions | 0.134(0.341) | 0.081(0.272) | 0.188(0.390) | 0.082(0.274) | 0.186(0.389) |
| Hospital quality | 3.519(0.878) | 3.479(0.851) | 3.557(0.902) | 3.497(0.857) | 3.539(0.897) |
| Doctor trust | 6.782(2.407) | 6.673(1.957) | 6.892(1.998) | 6.781(1.950) | 6.784(1.999) |
| Medical institution Satisfaction | 3.642(0.805) | 3.580(0.794) | 3.704(0.810) | 3.599(0.795) | 3.684(0.812) |
| Systemic healthcare issues perceptions | 6.658(2.737) | 7.177(2.457) | 6.139(2.900) | 7.167(2.486) | 6.156(2.878) |

**Table 4. Logit model estimation for factors influencing Internet use and Internet involvement.**

| Variables | Internet use | | | | Internet involvement | | | |
|---|---|---|---|---|---|---|---|---|
| | Coef. | Std. | z | P value | Coef. | Std. | z | P value |
| Gender | 0.036 | 0.040 | 0.90 | 0.368 | 0.004 | 0.038 | 0.11 | 0.908 |
| Age | -0.101 | 0.002 | -59.13 | 0.000 | -0.086 | 0.002 | -55.58 | 0.000 |
| Education level | 0.799 | 0.184 | 43.39 | 0.000 | 0.745 | 0.017 | 44.68 | 0.000 |
| Marital status | 0.072 | 0.029 | 2.48 | 0.013 | 0.033 | 0.028 | 1.19 | 0.233 |
| Urban or rural | 0.188 | 0.023 | 7.94 | 0.000 | 0.063 | 0.018 | 3.59 | 0.000 |
| Not lonely | 0.091 | 0.018 | 4.83 | 0.000 | 0.174 | 0.018 | 9.68 | 0.000 |
| Life satisfaction | -0.030 | 0.020 | -1.56 | 0.119 | 0.070 | 0.019 | 3.73 | 0.000 |
| Health status | -0.029 | 0.017 | -1.72 | 0.086 | -0.025 | 0.016 | -1.57 | 0.116 |
| Alcohol drinker | 0.027 | 0.052 | 0.52 | 0.605 | 0.102 | 0.498 | 2.04 | 0.041 |
| Chronic disease | 0.186 | 0.053 | 3.49 | 0.000 | 0.108 | 0.0501 | 2.13 | 0.033 |
| Medical insurance | 0.276 | 0.071 | 3.89 | 0.000 | 0.126 | 0.067 | 1.89 | 0.059 |
| Hospital admissions | -0.100 | 0.058 | -1.72 | 0.085 | -0.091 | 0.055 | -1.64 | 0.101 |
| Hospital quality | -0.054 | 0.021 | -2.54 | 0.011 | 0.009 | 0.020 | 0.49 | 0.624 |
| Observation | 25121 | | | | 25121 | | | |
| Prob>chi$^2$ | 0.000 | | | | 0.000 | | | |
| Pseudo R$^2$ | 0.4605 | | | | 0.4072 | | | |

the latter, as did healthcare-related variables, such as chronic disease, medical insurance, and drinking alcohol.

## Matching and balance test

This study employed several matching methods and finally selected nearest neighbor (within-caliper) matching for PSM, as it demonstrated the highest level of effectiveness. The caliper threshold was set to 0.05. A substitution method was then adopted to match as many treated samples as possible to the control samples. In the PSM for Internet use, 4 of 12563 user samples and 100 of 12558 non-user samples were not matched to an object, and 25107 samples were supported. Further, 51 of 12485 high-involvement samples and 93 of 12636 low-involvement samples were not matched, and 24977 samples were supported. The overall matching effect in this study was therefore sound.

A balance test was then conducted to evaluate the quality of matching and reliability of the estimated results. The results in the S1 Table show that the deviation between the control and treated groups reduced substantially after PSM matching. The standard deviations of all groups were within 10% after matching, indicating that the PSM effectively balanced the effects of the covariates on both groups.

## Sensitivity analysis

As Rosenbaum and Rubin proposed, sensitivity analysis is necessary for PSM [53]. The selectivity bias in non-randomized experiments can be reduced by matching, but there is still the possibility that confounding variables not included in the covariates may cause hidden bias. Sensitivity analysis is a good way to ensure that a hidden selection bias does not change the treatment effectiveness results. The gamma coefficient was used to represent the effect of undetected variables on Internet use and Internet involvement. S2 Table illustrates that even with an increased gamma coefficient of 3, the results retained their significance. This indicates that the findings successfully withstand the sensitivity test [53].

**Table 5. Effects of Internet use and involvement on attitudes to healthcare.**

| Variables | PSM | Treated | Control | difference | SE | T |
|---|---|---|---|---|---|---|
| | | | Internet use | | | |
| Doctor trust | Before PSM | 6.673 | 6.892 | -0.218 | 0.030 | -7.20*** |
| | After PSM | 6.673 | 6.628 | 0.045 | 0.112 | 0.40 |
| Satisfaction | Before PSM | 3.580 | 3.704 | -0.124 | 0.010 | -12.26*** |
| | After PSM | 3.580 | 3.648 | -0.068 | 0.036 | -1.85 |
| Systemic healthcare issue perception | Before PSM | 7.178 | 6.139 | 1.038 | 0.034 | 30.62*** |
| | After PSM | 7.178 | 6.796 | 0.382 | 0.123 | **3.10***** |
| | | | Internet involvement | | | |
| Doctor trust | Before PSM | 6.781 | 6.784 | -0.003 | 0.030 | -0.08 |
| | After PSM | 6.779 | 6.449 | 0.330 | 0.114 | **2.90**** |
| Satisfaction | Before PSM | 3.599 | 3.685 | -0.085 | 0.010 | -8.42*** |
| | After PSM | 3.599 | 3.673 | -0.074 | 0.035 | **-2.06*** |
| Systemic healthcare issue perception | Before PSM | 7.168 | 6.156 | 1.012 | 0.034 | 29.81*** |
| | After PSM | 7.166 | 6.561 | 0.605 | 0.124 | **4.88***** |

SE: standard error

*p<0.05, **p<0.01, ***p<0.001

## Final results

After propensity score matching, the study estimated the average treatment effect on the treated (ATT), as shown in Table 5. The table indicates that many results varied considerably before and after matching. Before matching, Internet use negatively affected doctor trust and medical institution satisfaction, and Internet involvement did not affect doctor trust. After matching, however, Internet use did not affect doctor trust and medical institution satisfaction at all, and Internet involvement increased doctor trust. This indicates that more accurate and balanced results can be obtained after eliminating confounding variables through PSM. The study also found that Internet involvement decreased medical institution satisfaction, and both Internet use and Internet involvement exaggerated the perception of systemic healthcare issues.

## Discussion

This study explored the effects of Internet use and involvement on multi-dimensional attitudes to healthcare, using the PSM method and data from CFPS 2018. The results found that both Internet use and involvement had an impact on residents' attitudes to healthcare in China, although they worked quite differently. Specifically: the former contributed to a rise in the level of systemic healthcare issue perception, and while the latter increased trust in doctors, it also impaired medical institution satisfaction, and exaggerated systemic healthcare issue perception, as shown in Fig 3.

This study makes several key theoretical contributions and provides powerful empirical evidence to explain the literature's inconsistent conclusions on related issues. First, the study explored attitudes to healthcare from three aspects: doctors, medical institutions, and systemic healthcare issues. It found that while simple Internet use did not impact residents' attitudes to doctor trust, Internet involvement had a positive effect. In line with media promotion theory [55], the study found that making full use of online health information can effectively reduce the information asymmetry between patients and doctors, and enhance doctor trust. Internet involvement reduces residents' satisfaction with healthcare organizations. Both Internet use

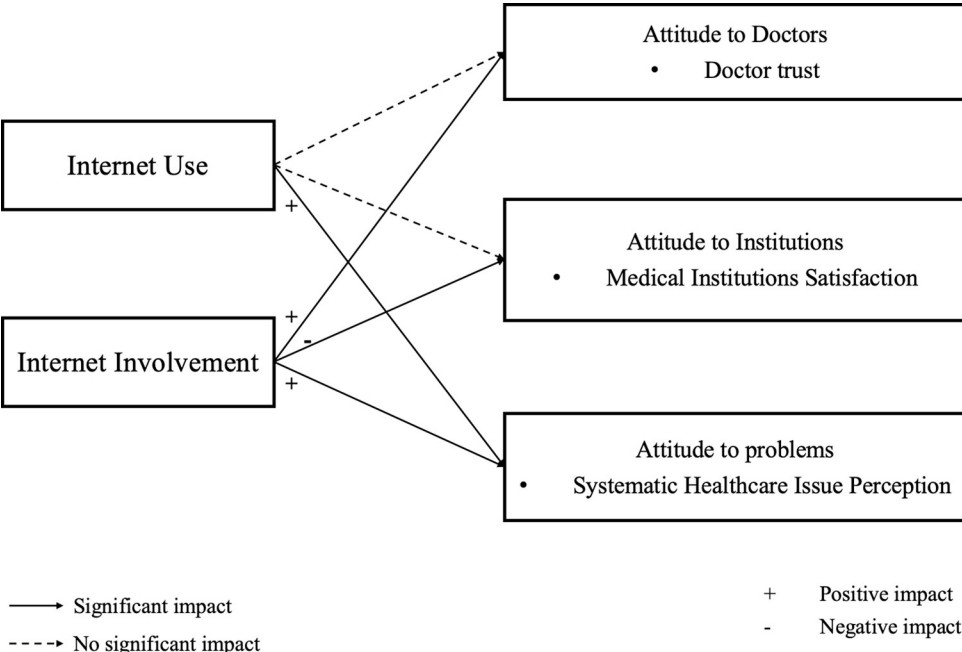

**Fig 3. The impact of Internet use and Internet involvement on attitudes toward healthcare.**

and involvement played a negative role in systemic healthcare issue perceptions in the study, which is consistent with media bias theory [35] and media depression theory [56]. The former theory suggests that distorting and filtering mechanisms can lead to bias among individuals. To pursue extra news value and cater to public psychology, media like the Internet tend to adopt a preferred viewpoint and narrative framework, and often present a negative image of medical institutions and systemic problems [57]. In some areas of China, there are many problems with the medical system (such as over-examination and corruption resulting from an information asymmetry between supply and demand), and the Internet provides a convenient platform for exposing these unsatisfactory medical institutions and incidents. Residents who frequently use the Internet to obtain health information are more likely to access such negative information, which in turn increases their anxiety [58].

This study reveals a distinction in how the Internet influences residents' attitudes to doctors, medical institutions, and systemic issues. This is probably because the increasing number of reports of incidents in which patients viciously attack doctors in China has led the public to show greater concern and respect for doctors and transfer their anger to medical institutions and systems. Doctors are no longer the only synonym for healthcare, and Chinese residents have begun to understand how important and difficult this occupation is. Attitudes to healthcare should therefore not be generalized in future research and should be discussed about different subjects and in different contexts. This also demonstrates that residents' access to and comprehension of different subjects within healthcare domains vary according to their level of Internet engagement.

Second, the study distinguished Internet use and Internet involvement, and explored the two mechanisms independently. The magnitude of Internet consumption, discrepancies in Internet access [59], and individual intentions have exerted an influence on information behavior [60]. Prior studies, however, generally measured Internet use as a dichotomy and ignored its extent and intention, potentially resulting in imprecise or conflicting outcomes. This study shows that simple use of the Internet, such as for purposes of entertainment or

social connection, is unlikely to affect residents' doctor trust and medical institution satisfaction because it is difficult for the public to access information about doctors or medical institutions unless they are seeking it. For those with higher Internet involvement, the Internet plays a much more important role in forming their attitudes to healthcare. On one hand, they are more likely to be exposed to medical information. Negative events such as over-medication and medical corruption are often at the center of media coverage and public opinions and therefore deepen misconceptions about healthcare institutions and systems. On the other hand, the Internet can contribute to health e-literacy and information literacy and allows the public to be more vigilant and critical when it encounters online health information. Internet involvement is therefore integral to research into the impact of the Internet on public attitudes to healthcare. This study complemented previous work by further exploring different types or degrees of Internet use, and attitudes to different aspects of healthcare in China.

Third, this study adopted a new statistical method to analyze the effects of Internet use and Internet involvement on residents' attitudes to healthcare. In the past, traditional quantitative research methods have not been strong enough to explain causality, and have lacked effective mechanisms to address endogeneity and heterogeneity. In particular, confounding covariates in secondary data could not be avoided. The results of previous research are therefore variable and unstable. This study used PSM to eliminate selection bias based on a counterfactual framework by matching and identifying the most confounding factors from the literature review. This analytic strategy generally supports the verification of causal relationships.

The study identified several key implications. Governments and medical institutions should optimize medical services and supervision, to change the public's relatively negative attitude from the source. They should then improve the publication and dissemination of medical information, to enhance its authority and ensure it is up-to-date. Internet media should replace the exaggerated reporting styles of the past with more rational and objective coverage, and focus on in-depth discussions rather than direct conclusions. The public's varied attitudes to doctors, medical institutions, and systemic healthcare problems reflect that the nature of Internet media reports can have a substantial impact on how individual opinions are formed. Online media should play a leading role in monitoring medical institutions and problems, and actively guide residents to develop an accurate view of the healthcare system. Finally, Chinese residents should improve their information literacy if they use the Internet frequently and intensively, so they can identify false or misleading information.

## Limitations

This study had some limitations, which need to be further explored. First, the thirteen confounding covariates in this study may not have included all possible factors. Since the data was from CFPS 2018, the variables in the secondary dataset were reasonably limited, and some covariates may have been ignored. Second, although this study reveals the relationship between Internet use and attitudes to healthcare, it is unable to explore the formation mechanism behind these relationships at the micro-cognitive level of individuals. In the future, this mechanism could be explored through primary data such as in-depth interviews, focus groups, and experiments. In addition, with the advancement of China's "Internet + Healthcare" strategy and the prevalence of the Internet, attitudes toward healthcare may vary over time. It is therefore necessary to consider these dynamic changes in future research.

## Conclusions

Attitudes to healthcare in China are closely tied to residents' health decisions and outcomes, influenced by the growing impact of the Internet in their daily lives. This study provided solid

empirical evidence for understanding the relationship between Internet use or involvement and attitudes to different aspects of healthcare, utilizing the PSM approach and the data from CFPS 2018. The findings showed that Internet involvement led to higher trust in doctors, lower satisfaction with medical institutions, and a heightened perception of systemic medical issues, while Internet use only increased the latter. This study unearthed the specific mechanism by which the Internet affects attitudes to healthcare and offers insights for Chinese governments, medical institutions, and residents.

## Supporting information

**S1 Table. Balance test results.**
(DOC)

**S2 Table. Sensitivity analysis.**
(DOC)

**S1 File. Data.**
(DTA)

## Acknowledgments

The authors extend their sincere thanks to Song Shijie for his assistance in this study.

## Author Contributions

**Conceptualization:** Xiaokang Song.

**Data curation:** Xinyue Li.

**Investigation:** Xinyue Li.

**Methodology:** Xinyue Li, Xiaokang Song.

**Project administration:** Xinyue Li.

**Resources:** Xiaokang Song.

**Validation:** Song Zhang.

**Writing – original draft:** Xinyue Li.

**Writing – review & editing:** Song Zhang, Xiaokang Song.

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
