## [Decision Letter · Decision Letter 0]

1 Aug 2023

PONE-D-23-10126The impact of Internet use and involvement on residents’ attitudes to healthcare in China: A propensity score matching analysisPLOS ONE

Dear Dr. Xiaokang,

Thank you for submitting your manuscript to PLOS ONE. After careful consideration, we feel that it has merit but does not fully meet PLOS ONE’s publication criteria as it currently stands. Therefore, we invite you to submit a revised version of the manuscript that addresses the points raised during the review process.

We look forward to receiving your revised manuscript.

Kind regards,

Chaohai Shen

Academic Editor

PLOS ONE

Journal Requirements:

"This work was supported by National Natural Science Foundation of China under Grant [number 72204210] and Scientific Research Foundation of Xuzhou Medical University under Grant [number D2021039]."

Reviewers' comments:

Reviewer's Responses to Questions

**Comments to the Author**

1. Is the manuscript technically sound, and do the data support the conclusions?

Reviewer #1: Partly

Reviewer #2: Yes

2. Has the statistical analysis been performed appropriately and rigorously? 

Reviewer #1: No

Reviewer #2: Yes

3. Have the authors made all data underlying the findings in their manuscript fully available?

Reviewer #1: Yes

Reviewer #2: Yes

4. Is the manuscript presented in an intelligible fashion and written in standard English?

Reviewer #1: Yes

Reviewer #2: Yes

5. Review Comments to the Author

Reviewer #1: The “Introduction” part is too lengthy. Most part of it consists of theoretical discussion. There should be a separate “Literature Review” Part. The paper should demonstrate an adequate understanding of the relevant literature in the field. There is a need to update recent literature up to 2023. The statistical analysis has to be performed rigorously. There are inconsistencies in result interpretations. The conclusion needs to tie together the other elements of the paper. The paper needs to be professionally written and its language should be smooth English.

Reviewer #2: The authors mainly investigated the impact of Internet use and involvement on residents’ attitudes to healthcare in China. This topic is interesting. But I have great concerns about this manuscript.

1. A sample selection diagram needs to be added to this manuscript so that readers can better understand the sample inclusion criteria.

2. There are many tables in the manuscript, and it is recommended to include some of the less important results as an appendix to the manuscript, such as a balance test.

3. The conclusion section currently has too many words and needs to be streamlined.

4. Authors need to change the abstract section to a structured abstract.

6. PLOS authors have the option to publish the peer review history of their article (what does this mean?). If published, this will include your full peer review and any attached files.

Reviewer #1: **Yes: **Sayed Farrukh Ahmed

Reviewer #2: **Yes: **Dai Su

---

## [Author Response · Author response to Decision Letter 0]

4 Jan 2024

Responses to the reviewers’ comments on the revised paper

“The impact of Internet use and involvement on residents’ attitudes to healthcare in China: A propensity score matching analysis”

Submitted to PLOS One

First and foremost, we would like to express our sincere gratitude to the reviewers for their valuable comments and constructive suggestions, which are all valuable and very helpful for revising and improving our paper. The paper is revised wherever are appropriate and possible according to these comments and suggestions. This report summarizes the revisions we made, in addition to our detailed responses to the comments and suggestions.We provided a file named "Responses to Reviewers" in attach files, please check. 

Reviewer 1

1. The “Introduction” part is too lengthy. Most part of it consists of theoretical discussion.

Reply: 

Thanks for your comments! We condensed the introduction section, now primarily encompassing two key segments. First, we introduced the background of rapid development of Internet in China, and the tight relationship between the Internet and healthcare services. Next, we introduced the motivation and objective of this research, highlighting the inconsistencies in existing literature regarding the impact of the Internet on the public’s attitudes towards healthcare. It is necessary to explore the true impact of Internet use to bolster its integration with healthcare services, particularly in near future and specifically in China. The manuscript was revised to include:

On pages 2 to 3: 

Introduction

China is one of the world’s largest online markets. The rapid development of the Internet and the inequality of the country’s urban and rural medical resources mean that increasing numbers of people regard the Internet as their main provider of health information. According to the 51st Statistical Report of the China Internet Network Information Center (CNNIC), by December 2022 the country boasted over than 64.66 million Internet users engaged in accessing healthcare-related information, accounting for 34.0% of all Internet users [1]. Emerging as a novel medium for the dissemination of health-related information, the internet assumes a pivotal role in addressing the scarcity of medical resources by delivering scientific knowledge and current news [2], but it also introduces inherent risks, such as misinformation. 

The Internet exerts a tangible influence on individuals’ attitudes to health, their decision making, and their relationship with the healthcare system [3-5]. Nonetheless, extant research has yielded inconsistent findings regarding the impact of the Internet on residents’ attitude toward healthcare. Some studies highlighted the potential benefits such as facilitating communication, and bridging the information asymmetry between doctors and patients, thereby improving doctor-patient relationships, and assisting individual health-related decision making [6]. On the other hand, massively misleading and misinterpreted information, the inconsistency of online sources with doctors’ diagnoses, and residents’ low health e-literacy could lead to doctor-patient conflict and the misunderstanding of healthcare information [7]. This is compounded by negative bias theory, which holds that individuals tend to pay more attention to negative than positive information [8-9]. The genuine impact of Internet on residents’ attitudes toward healthcare is undiscovered, especially in China. Comprehending this mechanism will facilitate a more effective utilization of the Internet’ s potential within the healthcare domain in near future. Therefore, this study aims to mitigate the incongruities in the conclusions drawn from aforementioned studies and to unearth the true impact of the Internet. Focusing on a more granular level of Internet use and Internet involvement, this study employs secondary data to explore the relationship between Internet and public attitudes toward healthcare in the Chinese context.

Additional references:

China Internet Information Center. The 51th CNNIC Report. 2023. https://www.cnnic.cn/NMediaFile/2023/0807/MAIN169137187130308PEDV637M.pdf. Accessed 8 August 2023.

Al-Haifi AR, Bumaryoum NY, Al-Awadhi BA, Alammar FA, Ashkanani RH, Al-Hazzaa HM. The influence of gender, media, and internet usage on adolescents' fast food perception and fluid intake. Journal of Health, Population and Nutrition. 2023;42(1):1-9.

2. There should be a separate “Literature Review” Part. The paper should demonstrate an adequate understanding of the relevant literature in the field. There is a need to update recent literature up to 2023.

Reply: 

Thanks for your comments! We added a new section entitled “Literature Review” into this study. This section comprises some theoretical discussion and literature review from the introduction in the previous version, while also incorporating additional new literature up to 2023. This section includes two main parts: Internet use and Internet involvement, Residents’ attitudes toward healthcare. More specifically, we initially integrated literature that delves into the effects of Internet use and involvement on public attitudes toward healthcare. Subsequently, we conducted a comprehensive review of the literature across three dimensions: attitudes toward doctors, medical institutions and issues. Further, we scrutinized the research method utilized in previous studies, pointing out that existing literatures often focused on the correlation rather than causal relationship. At last, we identified gaps in previous research and articulated our research objectives and proposed model. The corresponding revisions are as follow: 

On pages 3 to 5: 

Literature Review

Internet use and Internet involvement 

The impact of the Internet use on publics’ attitudes to healthcare has long been a concern. Previous studies mainly regarded Internet use as a simple concept, and only explored the impact of an individual’s decision of whether or not to use it on his or her attitude. For instance, Ybarra and Suman utilized secondary data from Surveying the Digital Future to investigate the potential impact of using the Internet to acquire health information on individuals’ healthcare experiences [6]. As DiMaggio and Hargittai stated, the dichotomous view as a distinction between people who do and do not have Internet access was conventionally regarded as a natural and customary standpoint in research [10]. Later, studies on the impact of media also found that different levels of media use could have different effects [11-12]. Internet use is a physical variable that describes whether or not an individual uses the internet, while Internet involvement is a psychological variable that describes how much an individual uses or values the Internet [13]. Yuan et al. subdivided Internet into Internet use and Internet involvement to understand the impact of Internet on the cognitive health of middle-aged [14]. Given the widespread use of the Internet and digital inequalities in China, Internet involvement is more likely to explain people’s attitudes and behaviors, rather than a mere binary of use or non-use. Therefore, research should move from the first level of the digital divide to the second, which addresses inequalities in levels and patterns of technology use [15].

Residents’ attitudes toward healthcare

When contemplating healthcare provisioning, a multiple of stakeholders come into play, encompassing entities such as doctors, regulators and health-related companies [16]. Because doctors are a common health information source for residents, trust in them plays an important role in the health decisions and outcomes of individuals [17-18]. Consequently, doctor trust emerges as a paramount concern. There are two conflicting viewpoints among scholars on the relationship between Internet use and doctor trust. Soroka et al. believed that the Internet’s flexibility and interactivity can promote mutual communication between doctors and patients, which in turn enhances doctor trust [19]. Other scholars, like Tan, held that people are easily attracted to negative information on the Internet, which damages doctor trust [20]. Pham et al. also found that when online health information contradicted a doctor’s opinion, individuals reported less trust in doctors and were more likely to seek suggestions from other sources [21]. 

Compared with their attitudes to doctors, people’s attitudes to medical institutions and problems are generally more negative and distrusting. A random sample survey of UK consumers found that the public generally showed low trust in healthcare institutions, but high trust in doctors [22]. Although the quality and coverage of online medical institutions have been improved in the context of China’s “Internet + Healthcare” strategy, Liu et al. found that Internet use may aggravate an individual’s dissatisfaction with medical services, which is consistent with the negative bias theory [23]. In addition, medical corruption, over-medication and drug unreliability are still inherent problems in China [24]. The complexity of these factors means that public attitudes to medical institutions and systemic healthcare issues remain under explored. 

There is an overall lack of empirical research into the relationship between different types of Internet use and multi-dimensional attitudes to healthcare. In addition, most existing literature researched the correlation (rather than the casual relationship) between Internet use and attitudes to healthcare, by adopting a statistical method such as regression analysis, and ignored the impact of many confounding factors [25]. To overcome the problems outlined previously, this study used propensity score matching (PSM) to explore the impact of Internet use and involvement on residents’ attitudes to healthcare in China. This study examined two mechanisms, as shown in Fig 1. The first focused on whether Internet use impacted attitudes to healthcare; and the second on the effect of Internet involvement. Internet involvement here refers to the significance of using the Internet as an information source, and can help obtain a better understanding of the specific effects of Internet use on attitudes to healthcare. The study divided public attitudes to healthcare into three categories: doctors, medical institutions and systemic healthcare issues.

Additional references:

Al-Haifi AR, Bumaryoum NY, Al-Awadhi BA, Alammar FA, Ashkanani RH, Al-Hazzaa HM. The influence of gender, media, and internet usage on adolescents' fast food perception and fluid intake. Journal of Health, Population and Nutrition. 2023;42(1):1-9.

Kim DKD, Kim S. What if you have a humanoid AI robot doctor?: An investigation of public trust in South Korea. Journal of Communication in Healthcare. 2022;15(4):276-85.

Liu H, Gong X, Zhang J. Does Internet Use Affect Individuals' Medical Service Satisfaction? Evidence from China. Healthcare. 2020;8(2):81. https://doi.org/10.3390/healthcare8020081 PMID: 32244464.

Yang X, Xin M, Liu K, Böke BN. The impact of internet use frequency on non-suicidal self injurious behavior and suicidal ideation among Chinese adolescents: an empirical study based on gender perspective. BMC public health. 2020;20(1):1-11.

3. The statistical analysis has to be performed rigorously.

Reply: 

Thanks for your comments! The statistical analysis in this study follows the established protocol of propensity score matching. We have refined the presentation of the data analysis with reference to previous studies using propensity score matching in PLOS One journal. Specifically, we first selected the samples (Fig 2) from CFPS dataset, and then screened the covariates. We have accordingly supplemented the figure of sample selection. And we added the demographics, baseline characteristics of respondents before matching (Table 2). We conducted the matching process (Table 3) and subsequent balance test (S1 Table), carried out sensitivity analyses (S2 Table) to corroborate the robustness of the outcomes. Finally, we computed the difference of three main dependent variables before and after PSM, and provided average treatment effect on the treated to derive this study’s conclusions (Table 4). The corresponding revisions are as follow:

On pages 12 to 15: 

Results

Descriptive statistics

Table 2 shows the respondents’ characteristics. This study used 25121 samples, among which the ratio of men to women was almost balanced. The average age was 47.213 years, with a range of 16 to 96. Most respondents’ education level was not high, most lived in rural areas, and most were married. Their health status was fair, and few had had a recent chronic disease. Over 90% of respondents had medical insurance. Their perceptions of happiness lack of loneliness were relatively high, with mean scores above 4. Respondents generally had a high level of trust in doctors, with a rating of 6.7 or above, but only a fair level of satisfaction with medical institutions, and high levels of concern about systemic healthcare issues in China. 

Propensity score

This study conducted logistic regression models to calculate respondents’ propensity scores for Internet use and Internet involvement, based on the thirteen covariates in Table 3. The logistic regression model is the model most frequently used to calculate propensity scores, since it does not contain requirements for normal distribution or type of covariates [32]. Overall, the explanatory power of the two models was relatively good, and many covariates influenced Internet use and Internet involvement. Specifically, demographic variables like age, education level, and living in rural or urban areas had statistical significance in both Internet use and Internet involvement. Loneliness and life satisfaction also had a significant impact on the latter, as did healthcare-related variables, such as chronic disease, medical insurance and drinking alcohol. 

Matching and balance test 

This study employed several matching methods, and finally selected nearest neighbor (within-caliper) matching for PSM, as it demonstrated the highest level of effectiveness. The caliper threshold was set to 0.05. A substitution method was then adopted to match as many treated samples as possible to the control samples. In the PSM for Internet use, 4 of 12563 user samples and 100 of 12558 non-user samples were not matched to an object, and 25107 samples were supported. Further, 51 of 12485 high-involvement samples and 93 of 12636 low-involvement samples were not matched, and 24977 samples were supported. The overall matching effect in this study was therefore sound. 

A balance test was then conducted to evaluate the quality of matching and reliability of the estimated results. The results in S1 Table show that the deviation between the control and treated groups reduced substantially after PSM matching. The standard deviations of all groups were within 10% after matching, indicating that the PSM effectively balanced the effects of the covariates on both groups. 

Sensitivity analysis

As Rosenbaum and Rubin proposed, sensitivity analysis is necessary in PSM [31]. The selectivity bias in non-randomized experiments can be reduced by matching, but there is still the possibility that confounding variables not included in the covariates may cause hidden bias. Sensitivity analysis is a good way to ensure that a hidden selection bias does not change the treatment effectiveness results. The gamma coefficient was used to represent the effect of undetected variables on Internet use and Internet involvement. S2 Table illustrates that even with an increased gamma coefficient of 3, the results retained their significance. This indicates that the findings successfully withstand the sensitivity test [31].

Final results

After propensity score matching, the study estimated the average treatment effect on the treated (ATT), as shown in Table 4. The table indicates that many results varied considerably before and after matching. Before matching, Internet use negatively affected doctor trust and medical institution satisfaction, and Internet involvement did not affect doctor trust. After matching, however, Internet use did not affect doctor trust and medical institution satisfaction at all, and Internet involv

---

## [Decision Letter · Decision Letter 1]

26 Mar 2024

PONE-D-23-10126R1The impact of Internet use and involvement on residents’ attitudes to healthcare in China: A propensity score matching analysisPLOS ONE

Dear Dr. Xiaokang,

Thank you for submitting your manuscript to PLOS ONE. After careful consideration, we feel that it has merit but does not fully meet PLOS ONE’s publication criteria as it currently stands. Therefore, we invite you to submit a revised version of the manuscript that addresses the points raised during the review process.

We look forward to receiving your revised manuscript.

Kind regards,

Chaohai Shen

Academic Editor

PLOS ONE

Additional Editor Comments:

Dear Authors,

I have received the required number of reviewer reports. Based on their comments and my own justification, I would like to invite you to revise your manuscript. Particularly, the theoretical analysis about the connection between the key variables should be important.

Sincerely,

Reviewers' comments:

Reviewer's Responses to Questions

**Comments to the Author**

1. If the authors have adequately addressed your comments raised in a previous round of review and you feel that this manuscript is now acceptable for publication, you may indicate that here to bypass the “Comments to the Author” section, enter your conflict of interest statement in the “Confidential to Editor” section, and submit your "Accept" recommendation.

Reviewer #1: All comments have been addressed

Reviewer #3: All comments have been addressed

2. Is the manuscript technically sound, and do the data support the conclusions?

Reviewer #1: Yes

Reviewer #3: No

3. Has the statistical analysis been performed appropriately and rigorously? 

Reviewer #1: Yes

Reviewer #3: No

4. Have the authors made all data underlying the findings in their manuscript fully available?

Reviewer #1: Yes

Reviewer #3: Yes

5. Is the manuscript presented in an intelligible fashion and written in standard English?

Reviewer #1: Yes

Reviewer #3: Yes

6. Review Comments to the Author

Reviewer #1: The paper has demonstrated an adequate understanding of the relevant literature in the field and cited an appropriate range of literature sources. The “Methods” part is well-written. The methods are employed appropriately and are able to meet the objectives of the study. The “Results” part is well-written. The conclusion adequately ties together the other elements of the paper.

Reviewer #3: 1. The author should further summarize the abstract of the article to highlight the research questions and research significance.

2. It is suggested to add part of the content to sort out the practical examples of the public obtaining medical resources through Internet channels. For example, how is the development of Internet medical treatment? What is the proportion of the public choosing online consultation or online consultation? How can the public obtain medical resources through the Internet in real life? What medical resources are obtain? The lack of reflection on the social reality problems will make the research meaningless.

3. Further differentiation between Internet use and Internet participation. The literature review section only gives a brief description of these two concepts and does not detail their differences, and Internet use and participation in health care security.

4. It is suggested to add a theoretical analysis section to analyze the connection between the key variables: Internet use and involvement, healthcare and public attitude, this can enable the theoretical basis of the research hypothesis.

7. PLOS authors have the option to publish the peer review history of their article (what does this mean?). If published, this will include your full peer review and any attached files.

Reviewer #1: **Yes: **Sayed Farrukh Ahmed

Reviewer #3: **Yes: **Shang Huping

---

## [Author Response · Author response to Decision Letter 1]

18 Apr 2024

For the editors:

1.The theoretical analysis about the connection between the key variables should be important.

Reply: Thanks for your comments! Theoretical analysis of the relationship between Internet usage/involvement and attitudes toward healthcare is crucial to this study. Therefore, we commenced by distinguishing between “Internet use” and “Internet involvement” in our literature (Table 1). This approach helps in understanding the significance and necessity of accurately defining the use of the Internet and delving into its impact on public’s attitudes. Next, we introduced a new section titled “Theoretical Analysis”. We discussed prevalent theories in related research, notably the media bias theory and the technology acceptance model. Despite these efforts, previous studies have shown inconsistent findings, likely due to not clearly distinguishing between the Internet use and Internet involvement, and a lack of detailed categorization of attitudes toward varied healthcare entities. Moreover, we propose that Internet use and Internet involvement have differential effects on the residents regarding various healthcare subjects. Specifically, different levels of Internet usage (use/involvement) empower the public in different ways and represent possibilities for accessing different healthcare subjects. Therefore, we deepened our discussions on the Internet’ impact on attitudes towards doctors, healthcare systems, and issues, drawing from a broad base of theoretical literature. Finally, we clearly defined the study’s focus and outlined a theoretical framework, as shown in Fig 1. 

For the reviewer 1

1. The paper has demonstrated an adequate understanding of the relevant literature in the field and cited an appropriate range of literature sources. The “Methods” part is well-written. The methods are employed appropriately and are able to meet the objectives of the study. The “Results” part is well-written. The conclusion adequately ties together the other elements of the paper.

Reply: Thanks for your valuable advice and acknowledgment! We are committed to refining the paper further and will persist in advancing our research in this domain in the future.

For the reviewer3:

1. The author should further summarize the abstract of the article to highlight the research questions and research significance.

Reply: Thanks for your comments! The abstract was structured into four segments: Background, Method, Results and Conclusions. Text within each segment was further streamlined for clarity and conciseness. In the Background, we emphasized the study’s research motivations and questions, focusing on resolving previous studies’ inconsistencies and uncovering the comprehensive mechanisms behind the Internet’s influence on residents’ multidimensional healthcare attitudes. In the Conclusions, we encapsulated the study’s significance both theoretically and practically. 

2. It is suggested to add part of the content to sort out the practical examples of the public obtaining medical resources through Internet channels. For example, how is the development of Internet medical treatment? What is the proportion of the public choosing online consultation or online consultation? How can the public obtain medical resources through the Internet in real life? What medical resources are obtain? The lack of reflection on the social reality problems will make the research meaningless.

Reply: 

Thanks for your comments! To more accurately reflect the current state of Internet healthcare development in China and to underscore the context and practical insights of this study, we reviewed numerous policy documents and industry development reports concerning Internet healthcare. This comprehensive review was added into the Introduction section in this revised manuscript.

Initially, we proposed the pressing need and imperative for the advancement of Internet healthcare in China. Given that China represents the world’s largest online market, the disparity and scarcity of healthcare resources have necessitated the Internet’s role as the primary conduit through which the public accesses health information. Subsequently, we illustrated the ways in which the public uses the Internet to access health resources (i.e., search engines, online health communities, Q&A forums, and social media) and the types of medical services they receive (i.e., online appointments, medications purchasing, remote consultations, disease management). Next, we explored the precise scope and services provided by Internet healthcare, employing detailed statistics that include the number of Internet hospitals, market dimensions, the range of services offered, the size of the user base, and other relevant data. This review allowed to present an objective and direct portrayal of the current state of the Internet healthcare. Given the growing prominence of the Internet within the healthcare domain, the urgency and necessity to explore the its impact on individual health attitudes and behaviors have intensified. 

3. Further differentiation between Internet use and Internet participation. The literature review section only gives a brief description of these two concepts and does not detail their differences, and Internet use and participation in health care security.

Reply:

Thanks for your comments! Our research centered on the nuanced distinctions between Internet use and Internet involvement, two pivotal concepts in understanding the relationship between online engagement and public attitudes toward healthcare. Internet use is defined as the binary act of accessing and utilizing the Internet primarily for information obtaining and decision-making. This can be simply measured by a “yes/no” - whether an individual uses the Internet or not. In contrast, Internet involvement delves deeper, capturing the significance and personal relevance attributed to the Internet by its users. This concept emerges from the proliferation and development of technology, portraying not just whether individuals access the Internet, but how essential it is to them and for what purposes they use it. To elucidate these distinctions for readers, we created Table 1, which presents a comparative analysis of these two concepts, enhancing the clarity of their unique aspects and interrelationships. 

Then, our review of literature explored the inconsistent impacts of Internet use on public attitudes toward healthcare. This variance among studies often stemmed from divergent interpretations of “Internet use”. Existing studies predominantly treated Internet use as a binary variable, neglecting the nuanced shifts in individual engagement and attitudes brought about by technological advancements and increased digital skills. In our comprehensive review of the literature focusing on Internet use and involvement, as well as public attitudes and behaviors within healthcare contexts, we identified a significant gap: the empirical investigation into the impact of Internet involvement is notably lacking. By clearly differentiating between Internet use and Internet involvement, we believed that we can more accurately gauge the effects of the Internet on individuals’ attitudes toward healthcare.

4. It is suggested to add a theoretical analysis section to analyze the connection between the key variables: Internet use and involvement, healthcare and public attitude, this can enable the theoretical basis of the research hypothesis.

Reply: Thanks for your comments! To enhance the elucidation of the correlation between Internet use/involvement and attitudes toward healthcare, and to furnish a theoretical framework for subsequent data analysis, we refined the section “Residents’ attitudes toward healthcare”, and introduced a new section titled “Theoretical Analysis”. 

In the section “Residents’ attitudes toward healthcare”, we explored current public attitudes towards healthcare and the factors influencing these perspectives across three dimensions: doctors, systems, and issues. In the new part titled “Theoretical Analysis”, we began by explaining the media bias theory and technology acceptance model, which we use to understand how the Internet influences public attitudes toward healthcare. This foundation highlights the challenges faced in reaching consistent conclusions across various studies, primarily due to a lack of specific differentiation between Internet use and Internet involvement, as well as overlooking the multifaceted nature of the attitudes under discussion. We proposed that varying levels of Internet usage (use/involvement) have differential effects on the residents regarding various healthcare subjects. Different levels of Internet usage empower the public in different ways and represent possibilities for accessing different healthcare subjects. Therefore, we delved into an extensive review of literature examining the Internet’s effect on attitudes toward doctors, healthcare systems, and issues. The research on the Internet’s impact on doctor trust was the most extensive yet shows significant inconsistencies in findings. Research on its influence on attitudes towards healthcare systems and issues is quite few, with conclusions tending towards a negative effect. Considering this gap in research and the findings from our literature review, we aimed to utilize propensity score matching to investigate the effects of Internet use and involvement on residents’ attitudes toward healthcare in China across three dimensions, as illustrated in Fig 1.

---

## [Decision Letter · Decision Letter 2]

3 May 2024

PONE-D-23-10126R2The impact of Internet use and involvement on residents’ attitudes to healthcare in China: A propensity score matching analysisPLOS ONE

Dear Dr. Xiaokang,

Thank you for submitting your manuscript to PLOS ONE. After careful consideration, we feel that it has merit but does not fully meet PLOS ONE’s publication criteria as it currently stands. Therefore, we invite you to submit a revised version of the manuscript that addresses the points raised during the review process.

We look forward to receiving your revised manuscript.

Kind regards,

Chaohai Shen

Academic Editor

PLOS ONE

Journal Requirements:

Additional Editor Comments:

Dear Authors,

I have received the reviewer report. The reviewer still has some concerns with your revised version, which I agree with. Please carefully read the report and make revisions accordingly.

Sincerely,

Reviewers' comments:

Reviewer's Responses to Questions

**Comments to the Author**

1. If the authors have adequately addressed your comments raised in a previous round of review and you feel that this manuscript is now acceptable for publication, you may indicate that here to bypass the “Comments to the Author” section, enter your conflict of interest statement in the “Confidential to Editor” section, and submit your "Accept" recommendation.

Reviewer #3: All comments have been addressed

2. Is the manuscript technically sound, and do the data support the conclusions?

Reviewer #3: Yes

3. Has the statistical analysis been performed appropriately and rigorously? 

Reviewer #3: Yes

4. Have the authors made all data underlying the findings in their manuscript fully available?

Reviewer #3: No

5. Is the manuscript presented in an intelligible fashion and written in standard English?

Reviewer #3: No

6. Review Comments to the Author

Reviewer #3: This version incorporates some of my suggestions, but there are still a few minor points that need to be addressed further. I will outline these minor points in my review below:

1.The abstract should be restructured and summarized concisely using standard academic English format. Integrate background, methods, results, and conclusions into one paragraph limited to 500 words rather than setting out each point separately.

2.The introduction and literature review can be streamlined, and it should be clearer about what exactly is new in the current study compared to existing literature.

3.The language should be proofread by native English speakers, and there are some typos in the manuscript. Please check it carefully before resubmitting.

4.Properly use citations and references in the manuscript, aiming for APA style or Harvard style. For example, (Baker et al., 2003) in the text, which could help readers to follow up on the references.

7. PLOS authors have the option to publish the peer review history of their article (what does this mean?). If published, this will include your full peer review and any attached files.

Reviewer #3: **Yes: **Shang Huping

---

## [Author Response · Author response to Decision Letter 2]

9 May 2024

Editor Comments:

The reviewer still has some concerns with your revised version, which I agree with. Please carefully read the report and make revisions accordingly.

Reply: Thanks for your comments! We have addressed the reviewer’s comments by revising each line accordingly. First, we consolidated and reconstructed the abstract into a single paragraph. We then streamlined the section of Introduction and Literature Review, and we further summarized the novelty of this study compared to other studies. This study differentiated between Internet use and Internet involvement, categorized public attitudes toward healthcare into different dimensions, and adopted innovative statistical methods to control for covariate selection bias. Additionally, we refined the text to ensure authenticity in English, corrected any typos, and thoroughly checked the references.

Reviewer 3:

1.The abstract should be restructured and summarized concisely using standard academic English format. Integrate background, methods, results, and conclusions into one paragraph limited to 500 words rather than setting out each point separately.

Reply: Thanks for your comments! We have revised the abstract by consolidating it into a single paragraph instead of listing each point separately. Initially, we provide an overview of the rapid development of the Internet within the healthcare domain and emphasize the importance of investigating its impact on residents’ healthcare attitudes. Subsequently, we highlight the research motivation, aiming to resolve inconsistencies and gaps in previous studies. We then outline our data sources and research method, followed by a brief presentation of our research findings. Finally, we underscore the significance of this study from both theoretical and practical perspectives.

2.The introduction and literature review can be streamlined, and it should be clearer about what exactly is new in the current study compared to existing literature.

Reply: Thanks for your comments! We streamlined the Introduction and Literature Review section by removing unnecessary details and enhancing coherence. In the concluding paragraph of the Introduction, we emphasize the unique contributions of this study compared to existing literature. These include the distinctions between Internet use and Internet involvement, the categorization of public attitudes toward healthcare, and the adoption of the research method and secondary dataset. Additionally, we also highlighted the research gap and our contributions in the concluding paragraph of each section within the Literature Review.

3.The language should be proofread by native English speakers, and there are some typos in the manuscript. Please check it carefully before resubmitting.

Reply: Thanks for your comments! The language in the whole article has been refined and optimized by native English speakers to improve the article’s flow and readability. In addition, a grammar checking tool has been utilized for double assurance. We have rectified any grammatical errors or typos that were present in the article. We are sorry for any misrepresentations and typos in the previous versions.

4.Properly use citations and references in the manuscript, aiming for APA style or Harvard style. For example, (Baker et al., 2003) in the text, which could help readers to follow up on the references.

Reply: Thanks for your comments! We reviewed the reference and made necessary revisions to rectify any inappropriate or incorrect parts. Regarding the reference format, we adhered to the author guidelines of PLOS ONE. We utilized the reference management tool Endnote and applied the style file provided by the PLOS ONE website. If there is a specific requirement to change to a different format, please let us know, and we will make the necessary adjustments accordingly.

---

## [Decision Letter · Decision Letter 3]

4 Jun 2024

The impact of Internet use and involvement on residents’ attitudes to healthcare in China: A propensity score matching analysis

PONE-D-23-10126R3

Dear Dr. Xiaokang,

We’re pleased to inform you that your manuscript has been judged scientifically suitable for publication and will be formally accepted for publication once it meets all outstanding technical requirements.

Kind regards,

Chaohai Shen

Academic Editor

PLOS ONE

Additional Editor Comments (optional):

Dear Authors,

The reviewer and I are satisfied with your revised version. I think it's good for publication.

Sincerely,

Reviewers' comments:

Reviewer's Responses to Questions

**Comments to the Author**

1. If the authors have adequately addressed your comments raised in a previous round of review and you feel that this manuscript is now acceptable for publication, you may indicate that here to bypass the “Comments to the Author” section, enter your conflict of interest statement in the “Confidential to Editor” section, and submit your "Accept" recommendation.

Reviewer #3: All comments have been addressed

2. Is the manuscript technically sound, and do the data support the conclusions?

Reviewer #3: Yes

3. Has the statistical analysis been performed appropriately and rigorously? 

Reviewer #3: Yes

4. Have the authors made all data underlying the findings in their manuscript fully available?

Reviewer #3: Yes

5. Is the manuscript presented in an intelligible fashion and written in standard English?

Reviewer #3: Yes

6. Review Comments to the Author

Reviewer #3: This revision takes into account my suggestions in a satisfactory way so I am happy to recommend publication.

7. PLOS authors have the option to publish the peer review history of their article (what does this mean?). If published, this will include your full peer review and any attached files.

Reviewer #3: **Yes: **Huping Shang

---

## [Editor Report · Acceptance letter]

1 Jul 2024

PONE-D-23-10126R3 

PLOS ONE

Dear Dr. Song, 

I'm pleased to inform you that your manuscript has been deemed suitable for publication in PLOS ONE. Congratulations! Your manuscript is now being handed over to our production team.

Kind regards, 

on behalf of

Dr. Chaohai Shen 

Academic Editor

PLOS ONE